# Electrochemical Disinfection of Experimentally Infected Teeth by Boron-Doped Diamond Electrode Treatment

**DOI:** 10.3390/jcm8122037

**Published:** 2019-11-21

**Authors:** Anna-Lena Böhm, Maximilian Koch, Stefan Rosiwal, Andreas Burkovski, Matthias Karl, Tanja Grobecker-Karl

**Affiliations:** 1Division of Ultra-Hard Coatings, Department of Material Sciences, University of Erlangen-Nuremberg, 91058 Erlangen, Germany; boehm.anna-lena@web.de; 2Microbiology Division, Department of Biology, University of Erlangen-Nuremberg, 91058 Erlangen, Germany; Maximilian.G.F.Koch@gmx.de (M.K.); Andreas.burkovski@fau.de (A.B.); 3Department of Material Sciences, University of Erlangen-Nuremberg, 91058 Erlangen, Germany; Stefan.Rosiwal@fau.de; 4Department of Prosthodontics, Saarland University, 66421 Homburg/Saar, Germany; tanja.grobecker-karl@uks.eu

**Keywords:** antimicrobial treatment, dentin tubules, electrochemical disinfection, endodontics, irrigating solutions, root canal

## Abstract

Disinfection and prevention of re-infection are the decisive treatment steps in endodontic therapy. In this study, boron-doped diamond (BDD) electrodes have been fabricated and used for disinfecting the root canals of extracted human teeth, which had been covered with bacterial biofilms formed by *Bacillus subtilis* and *Staphylococcus epidermidis*. The growth of *B. subtilis* could be successfully impaired, achieving a complete disinfection after 8.5 min treatment time with the success of disinfection depending on the insertion depth of the electrode in the root canal. *S. epidermidis* could completely be removed after 3.5 min treatment time. A clinically applicable electrode array led to complete disinfection after treatment times of 10 min for *S. epidermidis* and 25 min for *B. subtilis*. BDD electrode application allowed for the improved disinfection of root canals and dentin tubules based on a continuous production of reactive oxygen species and their enhanced penetration of dentin tubules most likely due the formation of a continuous stream of small gas bubbles. The treatment times that are required here will be shortened in clinical application, as mechanical shaping of the canal system would precede the disinfection process.

## 1. Introduction

Endodontic treatment is aimed at removing bacterial biofilm [1,2] from accessible canal space, proper disinfection [3,4], and prevention of reinfection by obturation and coronal restoration [5,6]. This approach is based on the assumption that apical periodontitis and bone resorption cannot develop in the absence of bacteria in the root canal system [7]. However, based on a large-scale clinical study showing only 67.4% of success [8], this cannot be predictably achieved.

One basic problem in disinfection is based on the fact that not all areas of the root canal system can be reached by irrigants [9,10], due to anatomic variations [2,11,12], apical size, and taper of canal preparation [13]. Furthermore, the effect of irrigating media is restricted to the root canal and the adjacent volume of dentin tubules [14], which only constitute a small proportion of free space available for bacterial growth inside a tooth [15]. Consequently, it has been shown that, within dentin canals, bacteria in established biofilms are less easily killed by endodontic medicaments than bacteria in developing, young biofilms [16]. Given that obturation material has been shown to either have no or quickly decreasing antimicrobial action [17], disinfection is required prior to obturation in order to avoid residual bacteria [6].

In the light of these problems, a huge body of literature exists regarding instrumentation techniques for chemo-mechanical preparation of root canals (for example, see [18,19,20,21]) and various irrigation protocols, including ultrasonic-assisted irrigation [22,23], which have been tested with at least partially contradictive results. Even advanced methods of disinfection, such as photodynamic therapy [24], laser-activated irrigation [25], or ozone application [26], as well as combined measures for disinfection, fail to completely eliminate bacteria from root canals [27]. 

In this study, we applied boron-doped diamond (BDD) electrodes for root canal disinfection. These electrodes have been successfully applied for electrochemical disinfection of sewage water, recently [28] and the antimicrobial effect is based on the unique features of the electrode material that require an overpotential of 2.8 V for anodic oxygen production in water, while disinfective OH·radicals are already generated at 2.5 V.

The goal of this study was to test the applicability of BDD electrodes in eliminating different bacterial biofilms from the root canal of experimentally infected human teeth.

## 2. Materials and Methods

### 2.1. Preparation and Infection of Extracted Teeth

The roots from carious-free extracted human teeth were obtained by horizontal cutting at the cementum-enamel junction while using a diamond band saw (EXAKT 300, EXAKT Advanced Technologies, Norderstedt, Germany). Root canals were subsequently prepared until reaching full apical patency while using standard reciprocating files (Reciproc, VDW, Munich, Germany) and conventional rinsing (sodium hypochlorite, chlorhexidine, ethanol). The prepared teeth were autoclaved for 20 min at 121 °C in distilled water and stored at ambient temperature until use. The sterilized teeth were transferred to Brain Heart Infusion (BHI) medium (Oxoid, Wesel, Germany) in test tubes freshly inoculated with *Staphylococcus epidermidis* DSM 20044 (German Type Culture Collection, DSMZ, Braunschweig, Germany) or *Bacillus subtilis* 168 (Bacillus Genetic Stock Centre, Columbus, OH, USA) and were subsequently incubated for three to five days at 30°C to induce colonization (Figure 1). *S. epidermidis* is a Gram-positive commensal bacterium of the skin and it was previously found in infected root canals together with *Bacillus* spec. [15]. *B. subtilis* was applied based on its ability to form highly resistant spores under the experimental conditions applied. 

For disinfection experiments, the teeth were removed from the test tubes and the outer surfaces were carefully sterilized while using a mixture of 32.5% isopropanol, 18% ethanol, 0.1% glutaraldehyde, and 49.4% distilled water, which is highly bactericidal [29] and routinely used for tooth sterilization by us (unpublished). The success of treatment was controlled by pressing the outer tooth surfaces on BHI agar plates, which were subsequently incubated overnight at 30 °C. Following individual placement in 1.5 mL reaction tubes, phosphate-buffered saline (137 mM NaCl, 2.7 mM KCl, 10 mM Na_2_HPO_4_, 2 mM KH_2_PO_4_, pH 7.4) was added as an electrolyte until the teeth were completely covered with liquid (Figure 2).

### 2.2. BDD Electrode Treatment Systems

Two different electrode arrays were used in this study. Prototype 1 consisted of two separate electrodes, which were positioned inside a root canal and inside the test tube, respectively (Figure 2). The depth of the root canal was measured and the electrode was inserted into the root canal only halfway for simulating a worst-case scenario of an insufficiently prepared or strongly ramified root canal and to evaluate the influence of insertion depth. In all experiments that were conducted with prototype 1 electrodes, the electric current applied ranged from 10 mA to 20 mA and from 1.5 V to 9.0 V.

Basically, allowing for clinical application, prototype 2 had both electrodes combined in one instrument, which could be inserted in a root canal (Figure 3). The electric current (6 V constant, 5 to 22 mA) was applied for up to 25 min.

For both prototypes, the BDD electrodes were produced by boron-doped diamond coating of a pure niobium wire with a diameter of approximately 0.2 mm in a hot-filament chemical vapor deposition plant under a gas atmosphere consisting of 98.4% hydrogen, 1.6% methane, and 0.005% trimethylborate. A uniform coating layer thickness of approximately 2 µm was achieved (Figure 4).

### 2.3. Analysis of Bacterial Growth

After treatment, the teeth were removed from the reaction tubes, longitudinally split [13,30] using a guillotine (Figure 5), impressed on BHI agar plates, and then incubated for 24 h at 37 °C. Bacterial growth was evaluated while using a scoring system (Figure 6), ranging from 1 to 4 (1 = no bacterial growth, 2 = moderate bacterial growth, 3 = medium bacterial growth, 4 = strong bacterial growth), depending on the level of bacterial growth seen on the plates.

## 3. Results

### 3.1. Removal of Biofilm from Root Canals Using a Two Wire BDD Electrode (Prototype 1)

While using prototype 1 electrodes, bacterial growth was impaired, depending on time and current. A complete disinfection of root canals colonized by *B. subtilis* was achieved after 8.5 min treatment time. In the case of *S. epidermidis*, complete disinfection was already observed after 3.5 min treatment, depending on the current applied (Figure 7).

### 3.2. Influence of Depth of Electrode Insertion on Disinfection Success

The electrode was inserted to different depth in prepared teeth colonized by *B. subtilis* biofilm to mimic an insufficiently prepared or strongly ramified root canal. In this set of experiments, the success of disinfection was not only depending time and applied current, but also on the insertion depth of the electrode in the root canal (Figure 8). Complete insertion led to a faster elimination of *B. subtilis*.

### 3.3. Application of Prototype 2 BDD Electrode Suitable for Clinical Application 

Prototype 1, which consisted of two separate electrodes and was used as a proof-of-principle device, worked sucessfully, but handling was inconvenient and not suitable for clinical application. Therefore, a double electrode device compromising only one distinct tip for disinfection was designed and tested.

For a better resolution of the disinfection process, for each time point, split teeth were impressed 10 times on BHI agar plates and growth was monitored after overnight incubation at 37 °C. When the prototype 2 electrode was applied, *B. subtilis* was completely eliminated within 25 min, while the complete elimination of *S. epidermidis* was already achieved after 10 min (Figure 9).

## 4. Discussion

Two different arrays of boron-doped diamond electrodes for the disinfection of root canals were tested. A clear dependency of the disinfection effectiveness on the charge quantity applied was found with *B. subtilis* showing greater resistance than *S. epidermidis*. With prototype 2 electrodes already forming a clinically applicable instrument, the method used here seems to be suitable for treating even robust bacterial contaminations. As compared to standard treatment with non-activated irrigating solutions [14], disinfection by BDD electrode application allowed for the improved sterilization of dentin tubules. This can be explained by a continuous production of reactive oxygen species and their enhanced penetration of dentin tubules due to the formation of a constant stream of gas bubbles resulting in a liquid flow within the free tooth volume.

The mechanical shaping of root canals [18,19,20,21] was done prior to biofilm formation and disinfection. As has been shown, the complete insertion of the electrode into the root canal is required for achieving adequate disinfection. Hence, in a clinical setting, BDD disinfection would have to be carried out as a final step of endodontic treatment prior to root canal obturation. As applying higher voltage would rather result in the formation of oxygen than in the formation of reactive oxygen species, charge quantity, which is critical for disinfection efficiency, can only be increased by longer treatment times. While the maximum treatment times that were applied here might be considered as being too long from a clinical perspective, it has to be kept in mind that, other than in the experiment conducted here, highly infected dentin would already be removed by mechanical shaping.

A wide variety of bacteria have been found in the root canals of infected teeth, out of which only two were chosen for the experiments described here. Apart from advantages in handling of the microorganisms, *S. epidermidis* was selected as example for a bacterium already found in root canals [15] and model for other root canal colonizing members of the genus [31]. *B. subtilis* was chosen as a very robust Gram-positive bacterium and model for other *Bacillus* species and Gram-positives found in root canals [15,31]. The higher resistance against BDD electrode treatment that was observed for *B. subtilis* might be explained by spore formation. Under the growth condition used, the *B. subtilis* cultures already started sporulation. These spores are highly resistant against physical and chemical stress, including reactive oxygen species. Even in case of spores, the treatment regimen applied led to complete eradication within a few minutes.

Furthermore, the application of a specific irrigation protocol during initial root canal preparation might be seen as a limitation of this experiment. It might have been advantageous to use EDTA for opening dentinal tubules to allow for easier penetration of bacteria. EDTA was not used here in order to stay in line with previous investigations [14] showing bacterial penetration into dentine using this irrigation protocol.

Despite the promising results obtained in this in vitro pilot study, comparative studies using accepted treatment strategies as controls as well as a wider variety of microorganisms are required. These had not been performed as a part of this investigation, which was designed as a proof-of-principle study for the electrochemical disinfection of root canals. Additionally, the potentially negative side effects of electrochemical disinfection on the periapical area require evaluation in animal models prior to clinical application.

## 5. Patents

Stefan Rosiwal, Andreas Burkovski and Matthias Karl have a filed a patent for the disinfection method described in this report.

## Figures and Tables

**Figure 1 jcm-08-02037-f001:**
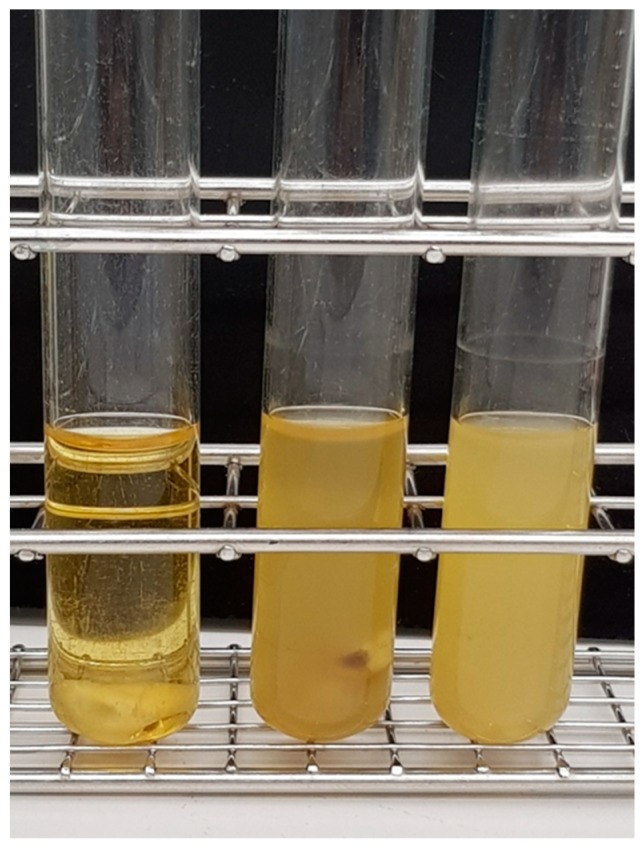
Colonization of tooth specimen. Teeth were placed in test tubes containing Brain Heart Infusion (BHI) medium inoculated with either *B. subtilis* or *S. epidermidis* and incubated for at least three days at 30 °C to allow for colonization and biofilm development. From left to right: inoculated test tube before incubation, *B. subtilis* and *S. epidermidis* culture after one day of bacterial growth.

**Figure 2 jcm-08-02037-f002:**
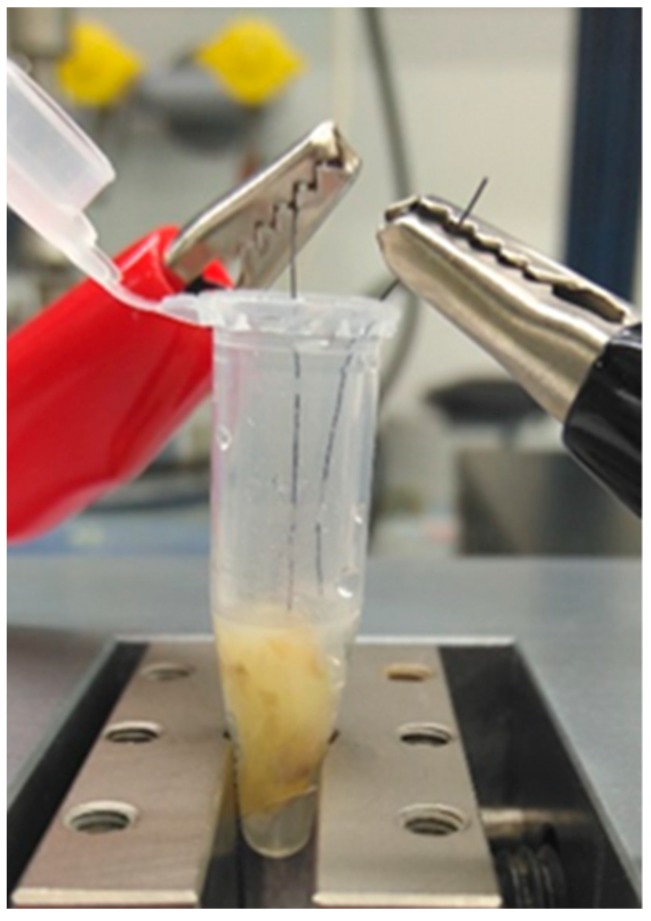
Experimental set-up for disinfection experiments using prototype 1 electrodes. The teeth were placed in 1.5 mL reaction tubes and phosphate-buffered saline (PBS) was added until the tooth was completely covered with liquid. The two separate electrodes were positioned inside the root canal and inside the tube, respectively.

**Figure 3 jcm-08-02037-f003:**
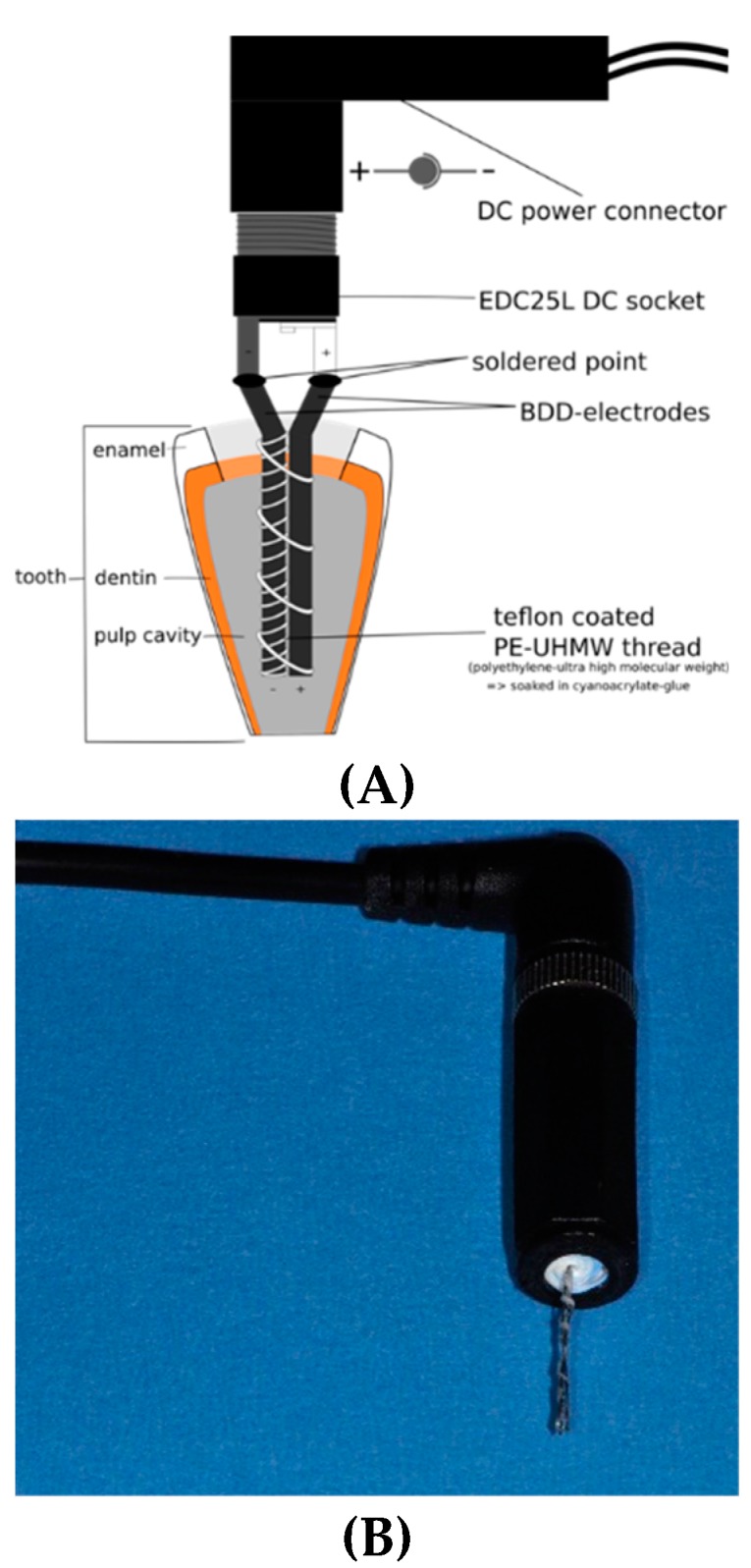
Prototype 2 boron-doped diamond (BDD) electrode used in this study, which can be inserted into a root canal (**A**): schematic representation; (**B**): actual prototype.

**Figure 4 jcm-08-02037-f004:**
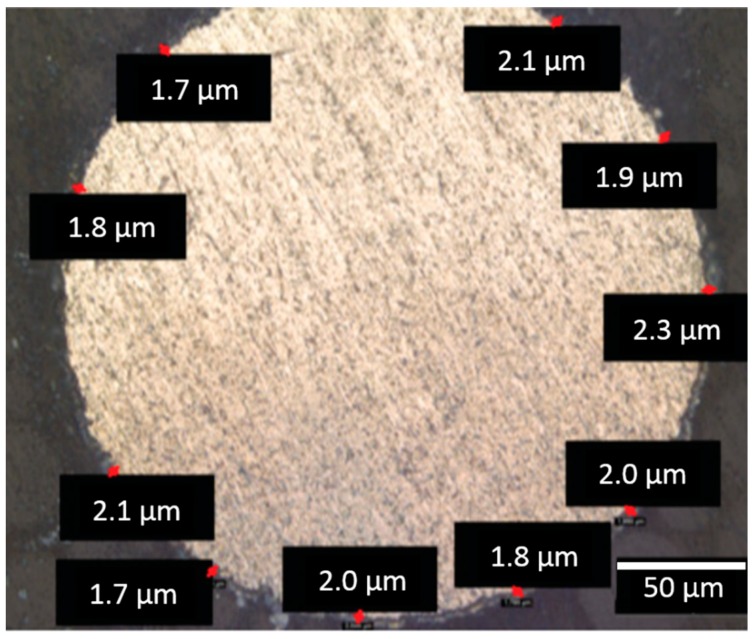
Scanning electron microscopy of BDD-covered niobium wire in cross section. As indicated by the individual measurements (red marks), a uniform coating thickness of approximately 2 µm was achieved (scale bar = 50 µm).

**Figure 5 jcm-08-02037-f005:**
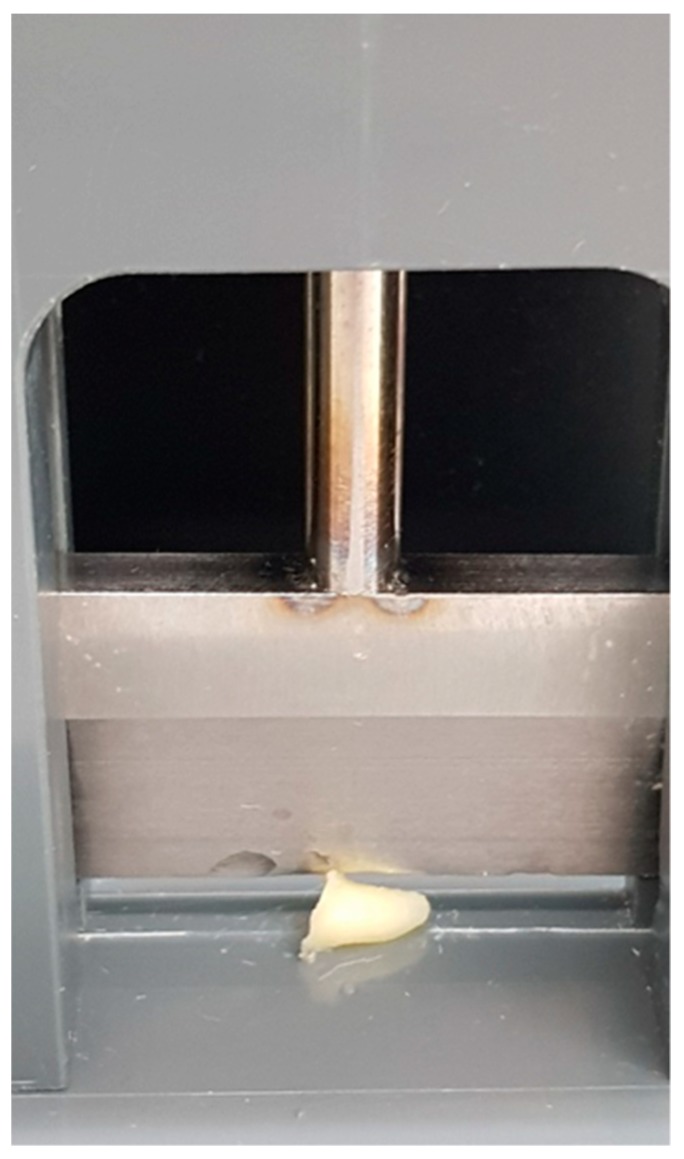
Guillotine used to longitudinally split treated teeth.

**Figure 6 jcm-08-02037-f006:**
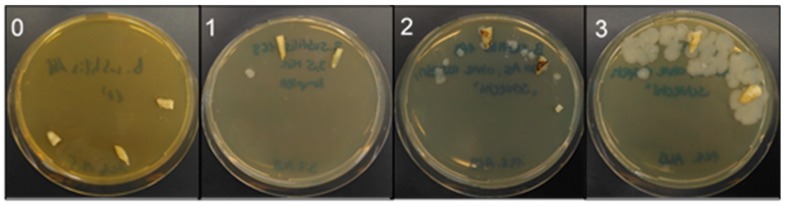
Evaluation scheme for bacterial growth following BDD electrode treatment. Split teeth were impressed on BHI agar plates and incubated overnight at 30 °C. Bacterial growth was scored from 0 = no bacterial growth, 1 = moderate bacterial growth, 2 = medium bacterial growth, and 3 = strong bacterial growth.

**Figure 7 jcm-08-02037-f007:**
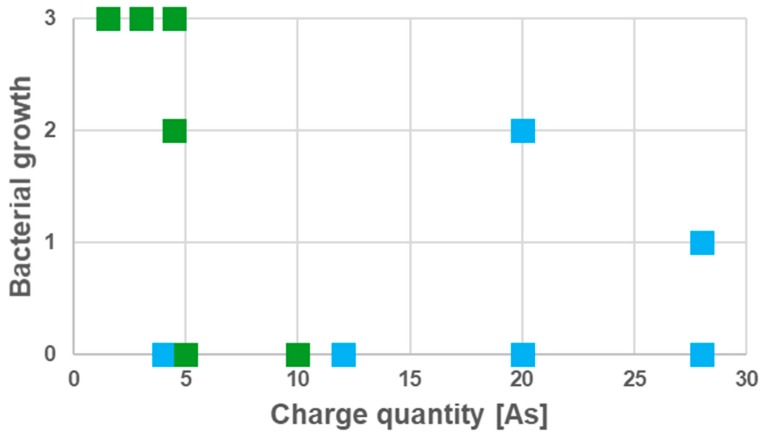
Effect of BDD electrode treatment on Gram-positive bacteria. Elimination of microorganisms depending on the applied charge quantity is shown and currents between 10 and 20 mA were applied in the different experiments (see Figure 6 for evaluation scheme; color coding: *B. subtilis*: green, *S. epidermidis*: blue, every square corresponds to an independent set of experiments, i.e., *n* = 6 for each species).

**Figure 8 jcm-08-02037-f008:**
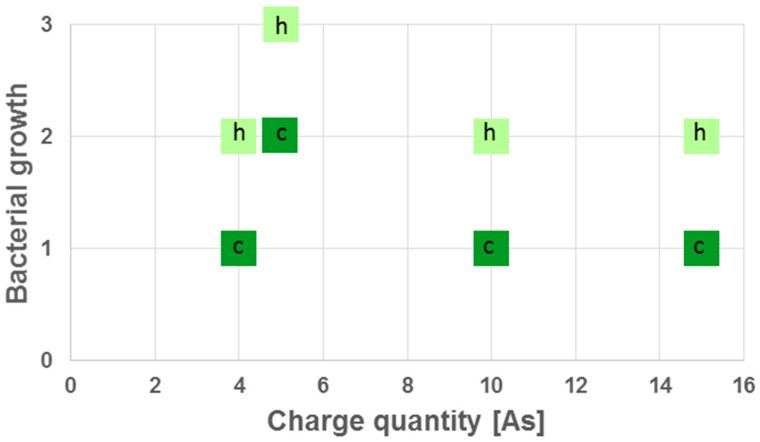
Effect of electrode insertion depth on success of disinfection. Growth of *B. subtilis* dependent on applied charge quantity and insertion depth of the electrode was tested (h = halfway, c = complete; see Figure 6 for evaluation scheme).

**Figure 9 jcm-08-02037-f009:**
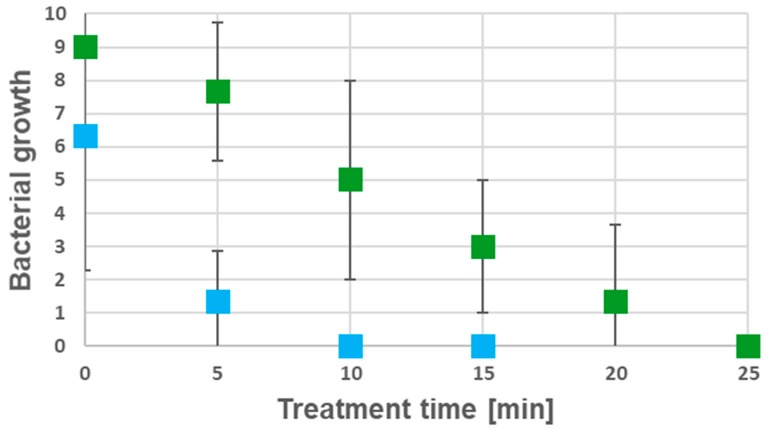
Disinfection kinetics of prototype 2 electrodes. Growth of *B. subtilis* and *S. epidermidis* dependent on treatment time is shown (bacterial growth is indicated as numbers of contaminated imprints showing bacterial growth after overnight incubation; color coding: *B. subtilis*: green, *S. epidermidis*: blue.) Each set of experiments was carried out in three independent biological replicates (*n* = 3) and each square shows the mean of the corresponding data (± standard deviation).

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
