# Peer review of "Electrochemical Disinfection of Experimentally Infected Teeth by Boron-Doped Diamond Electrode Treatment"

_jcm, 2019, doi:10.3390/jcm8122037_

Round 1

Reviewer 1 Report

I think, that this work is well planed and done, with adequately described methods, results, discussion and conclusions.
The paper "Electrochemical disinfection of experimentally infected teeth by boron-doped diamond electrode treatment" has an informative abstract. The manuscript starts with detailed review of the endodontic therapeutic problem in the “Introduction”. The “Material and Methods” section is an exhaustive presentation of to the modern techniques used in this study. In the parts “Results" and "Discussion” are presented clearly and informative the successful effect of studied electrochemical BDD disinfection. A stronger effect was observed against Staphylococcus epidermidis and less effective against Bacillus subtilis, as sporoforming bacilli have a higher resistance to disinfection by any technique. The strong bactericidal in vitro effect and potentially negative side effects of the electrochemical disinfection on the periapical area requare in vivo evaluation in other study.

Author Response

Response to Reviewer #1

Thank you very much!

We fully agree, that the study conducted is only a starting point and we have added a statement to the Discussion, which now reads, “Also, potentially negative side effects of electrochemical disinfection on the periapical area require evaluation in animal models prior to clinical application.”

Reviewer 2 Report

 Page 2 line 55. Please use a better LAI reference. this study is not appropriate because the authors did not disclose several aspects of their methodology and they did not used NaOCl. You need to support with stronger evidence what could be the real difference in between your proposed method to LAI.

Page 2 Line 70. Why did you used CHX and ethanol during the root canal instrumentation? Teeth should have been irrigated with NaOCl and EDTA for the opening of the dentinal tubules for the better infection of the root canal and dentinal tubules. NaOCl needs to be inactivated with sodium thiosulfate (Na2S203) previously to be autoclaved

Line 73-74. Why did authors decided to used S epidermidis and B subtilis?

Page 3 line 83. Please include the reference for the followed disinfection protocol and provide a solid evidence on  why you followed this specific protocol. 

Page 3 line 92. Please explain the reason why electrodes where place halfway or completely inside the root canal and how did you determined the difference according to your results?

Page 3 line 95. Please include apart from the diagram  a picture of the prototype 2. Or this is not possible due Patent reasons?

Page 4 line 114. Please explain how did you prevent your microbiology sampling from getting contaminated (external tooth surface decontamination)

Page 4 line 116. The score table is useful,  but serial dilutions should have been made to obtained more specific results.

Page 8 line 193. You should have included 2 control groups (water and NaOCl) to compared your findings.

What make this new electrode device proposal different from Endox? (Virtej et al. JOE;33(8) 2007

Author Response

Response to Reviewer #2

English language and style are fine/minor spell check required

We once again went through the manuscript checking for style and language errors

Page 2 line 55. Please use a better LAI reference. this study is not appropriate because the authors did not disclose several aspects of their methodology and they did not used NaOCl. You need to support with stronger evidence what could be the real difference in between your proposed method to LAI.

We have replaced reference #25 with Gokturk et al. 2017

Page 2 Line 70. Why did you used CHX and ethanol during the root canal instrumentation? Teeth should have been irrigated with NaOCl and EDTA for the opening of the dentinal tubules for the better infection of the root canal and dentinal tubules. NaOCl needs to be inactivated with sodium thiosulfate (Na2S203) previously to be autoclaved

We have added a comment in the Discussion, which reads, “Furthermore, the application of a specific irrigation protocol during initial root canal preparation may be seen as a limitation of this experiment. It might have been advantageous to use EDTA for opening dentinal tubules in order to allow for easier penetration of bacteria. In order to stay in line with previous investigations showing bacterial penetration into dentine using this irrigation protocol, EDTA was not used here.”

Line 73-74. Why did authors decided to used S epidermidis and B subtilis?

We have added an explanation, which reads “S. epidermidis is a Gram-positive commensal bacterium of the skin and was previously found in infected root canals together with Bacillus spec. [15]. B. subtilis was applied based on its ability to form highly resistant spores under the experimental conditions applied.”

Page 3 line 83. Please include the reference for the followed disinfection protocol and provide a solid evidence on why you followed this specific protocol.

We have added an explanation, which reads “For disinfection experiments, the teeth were removed from the test tubes and the outer surfaces were carefully sterilized using a mixture of 32.5% isopropanol, 18% ethanol, 0.1% glutaraldehyde and 49.4% distilled water, which is highly bactericidal [Karsa 2007] and routinely used for teeth sterilization by us (unpublished). Success of treatment was controlled by pressing the teeth surfaces on BHI agar plates, which were subsequently incubated overnight at 30°C (data not shown).”

Page 3 line 92. Please explain the reason why electrodes where place halfway or completely inside the root canal and how did you determined the difference according to your results?

We have further elaborated on the explanation already included and the respective section now reads “For simulating a worst-case scenario of an insufficiently prepared or strongly ramified root canal and to evaluate the influence of insertion depth, the depth of the root canal was measured and the electrode was inserted into the root canal only halfway.”

Page 3 line 95. Please include apart from the diagram a picture of the prototype 2. Or this is not possible due Patent reasons?

We have added a picture showing the prototype 2 electrode

Page 4 line 114. Please explain how did you prevent your microbiology sampling from getting contaminated (external tooth surface decontamination)

The external surface decontamination has now been described in detail and a reference (Karsa 2007) has been added - Please cf. your comment on Page 3 line 83.

Page 4 line 116. The score table is useful, but serial dilutions should have been made to obtained more specific results.

Serial dilutions were not possible, since every time point studied is represented by a single tooth, which was split and pressed on BHI agar plates with its inner surface (see Figure 6).

Page 8 line 193. You should have included 2 control groups (water and NaOCl) to compared your findings.

We fully agree with the reviewer that additional comparative testing with established methods has to be carried out and we have further stressed this point in the Discussion which now reads “Despite these promising results obtained in this in vitro pilot study, comparative studies using accepted treatment strategies as controls as well as a wider variety of microorganisms are required. These had not been performed as a part of this investigation which was designed as a proof-of-principle study for electrochemical disinfection of root canals.”

What make this new electrode device proposal different from Endox? (Virtej et al. JOE;33(8) 2007

We are aware of the Endox instrument which acts similar to an electrotome. We have not mentioned Endox as to our knowledge it never came into widespread use and the mode of action is completely different. As briefly stated in the Introduction, BDD electrodes so far are the only known electrode material capable of directly producing disinfective radicals from water. If desired, we would be willing to include the reference mentioned but are afraid that this might confuse the reader.

Round 2

Reviewer 2 Report

Thank you.